# The Developmental Implications of Muscle-Targeted Magnetic Mitohormesis: A Human Health and Longevity Perspective

**DOI:** 10.3390/bioengineering10080956

**Published:** 2023-08-12

**Authors:** Alfredo Franco-Obregón, Yee Kit Tai, Kwan Yu Wu, Jan Nikolas Iversen, Craig Jun Kit Wong

**Affiliations:** 1Department of Surgery, Yong Loo Lin School of Medicine, National University of Singapore, Singapore 119228, Singapore; lesleywu77@live.nl (K.Y.W.); nikolas.iversen@u.nus.edu (J.N.I.); surwjkc@nus.edu.sg (C.J.K.W.); 2Institute of Health Technology and Innovation (iHealthtech), National University of Singapore, Singapore 117599, Singapore; 3Biolonic Currents Electromagnetic Pulsing Systems Laboratory (BICEPS), National University of Singapore, Singapore 117599, Singapore; 4NUS Centre for Cancer Research, Yong Loo Lin School of Medicine, National University of Singapore, Singapore 117599, Singapore; 5Department of Physiology, Yong Loo Lin School of Medicine, National University of Singapore, Singapore 117593, Singapore; 6Healthy Longevity Translational Research Programme, Yong Loo Lin School of Medicine, National University of Singapore, Singapore 119228, Singapore; 7Nanomedicine Translational Research Programme, Centre for NanoMedicine, Yong Loo Lin School of Medicine, National University of Singapore, Singapore 117544, Singapore; 8Faculty of Medicine, Utrecht University, 3584 CS Utrecht, The Netherlands

**Keywords:** healthspan, PEMF, magnetic fields, muscle secretome, exercise

## Abstract

Muscle function reflects muscular mitochondrial status, which, in turn, is an adaptive response to physical activity, representing improvements in energy production for de novo biosynthesis or metabolic efficiency. Differences in muscle performance are manifestations of the expression of distinct contractile-protein isoforms and of mitochondrial-energy substrate utilization. Powerful contractures require immediate energy production from carbohydrates outside the mitochondria that exhaust rapidly. Sustained muscle contractions require aerobic energy production from fatty acids by the mitochondria that is slower and produces less force. These two patterns of muscle force generation are broadly classified as glycolytic or oxidative, respectively, and require disparate levels of increased contractile or mitochondrial protein production, respectively, to be effectively executed. Glycolytic muscle, hence, tends towards fibre hypertrophy, whereas oxidative fibres are more disposed towards increased mitochondrial content and efficiency, rather than hypertrophy. Although developmentally predetermined muscle classes exist, a degree of functional plasticity persists across all muscles post-birth that can be modulated by exercise and generally results in an increase in the oxidative character of muscle. Oxidative muscle is most strongly correlated with organismal metabolic balance and longevity because of the propensity of oxidative muscle for fatty-acid oxidation and associated anti-inflammatory ramifications which occur at the expense of glycolytic-muscle development and hypertrophy. This muscle-class size disparity is often at odds with common expectations that muscle mass should scale positively with improved health and longevity. Brief magnetic-field activation of the muscle mitochondrial pool has been shown to recapitulate key aspects of the oxidative-muscle phenotype with similar metabolic hallmarks. This review discusses the common genetic cascades invoked by endurance exercise and magnetic-field therapy and the potential physiological differences with regards to human health and longevity. Future human studies examining the physiological consequences of magnetic-field therapy are warranted.

## 1. Introduction

Muscle is our largest tissue mass and, as such, has evolved to modulate the regenerative capacity and metabolism of the entire body. To fulfil this role, muscle acts as a feeding tissue for the rest of the body. This feeding role of muscle is a manifestation of its secretome, a vast combination of regenerative, metabolic, anti-inflammatory and immunocompetence factors, released into the systemic circulation as either individual myokines (muscle-derived cytokines) [1,2,3] or vesicle-encapsulated factors [4,5,6]. Mechanistically, an upregulation in mitochondrial respiratory rate triggers the myokine response pathway [7,8,9,10], whereas extracellular calcium entry [11] as well as mitochondrial respiration [7] stimulate the release of muscular extracellular vesicles. Physical activity, or exercise, is the most common way to initiate the myokine [12,13] and extracellular vesicle [6] responses, which are, hence, blunted in the old and frail who are less capable of undertaking exercise [14,15,16]. Upon elevated muscle mitochondrial respiration induced by exercise, components of the muscle secretome travel via the bloodstream to the collateral tissues of the body that, in turn, may reciprocate with secretome responses of their own. Collateral tissue systems known to respond to the actions of the muscle secretome include the bone and joints, the immune system, the central nervous system, the digestive system, the microbiome, and, in particular, adipose tissue [3]. The muscle secretome also acts in an autocrine manner, whereby muscle itself is the target of its own secretome to produce metabolic and regenerative adaptations. A system of endocrine/paracrine/autocrine cross-talk, hence, ensues, that adapts the body’s regenerative and metabolic statuses [17]. 

## 2. Muscle Metabolic Phenotypes

Oxidative muscles display a predilection for fatty-acid oxidation to provide energy, whereas glycolytic muscles rely largely on carbohydrates for energy production. Oxidative- and glycolytic-muscle phenotypes subserve distinct forms of contractile activity, which mirrors these distinct modes of energy production. Endurance exercise, such as distance running, predominantly recruits the participation of oxidative muscles. Oxidative muscles mediate tonic muscle contractures that require sustained mitochondrial aerobic energy production, predominantly from fatty acids. Consequently, force generation from oxidative muscle is slower and produces less power. On the other hand, resistance exercise, such as weight lifting, largely relies on the participation of glycolytic muscles. Glycolytic muscles mediate sudden bursts of contraction, requiring immediate, albeit relatively short-lived, energy production from carbohydrates, without the assistance of oxygen (anaerobically) outside of the mitochondria. Consequently, glycolytic muscles produce more powerful contractures than oxidative muscles, but exhaust more rapidly. Glycolytic muscles also accrue a substantial oxygen debt in anticipation of final mitochondrial oxidation of preliminarily catalysed carbohydrates in the form of lactic acid [18]. Extending from their distinct contractile signatures, oxidative and glycolytic muscles have also earned the names slow-twitch (type I) and fast-twitch (type II) muscles, respectively (Figure 1). Therefore, compared to glycolytic muscles, oxidative muscles exhibit greater mitochondrial numbers, have a higher reliance on oxidative metabolism [19] and are better able to support systemic insulin-sensitivity due to their predilection for fatty-acid oxidation [20]. Two transcriptional cascades predominantly determine the oxidative phenotype, calcium- or mitochondria-mediated.

## 3. Myoplasmic Calcium Levels

The oxidative-muscle phenotype is characterised by elevated resting calcium levels (100–300 nM), whereas glycolytic muscles are distinguished by having lower basal calcium levels (<50 nM), interspersed by high amplitude calcium transients [21,22]. These distinct calcium signatures are largely established by voltage-gated calcium entry, downstream of neuronal stimulation, and the ensuing calcium-mediated excitation–contraction coupling mechanism. In combination, these two calcium pathways serve as a developmental directive for the determination of embryonic muscle types or for the adaptive remodelling of the metabolic and functional properties of existing muscles. Mechanical loading also contributes to resting calcium levels, particularly within the functional context of oxidative muscles [21,22]. 

Calcium serves an enzyme catalytic role in skeletal-muscle determination. The calcium-dependent phosphatase, calcineurin (protein phosphatase 2B), is preferentially activated by the sustained calcium levels that characterise oxidative muscle [21,22,23,24]. Activated calcineurin, in turn, dephosphorylates the nuclear factor of activated T cells (NFAT), thereby allowing it to be translocated into the nucleus where it acts in cohort with other transcription factors to promote the oxidative-muscle phenotype in a calcium-sensitive manner [25]. Accordingly, specifically driving muscular calcineurin activity recapitulates the fast-to-slow muscle fibre switch [26,27], whereas calcineurin genetic knockdown or pharmacological inhibition of calcineurin decreases oxidative-muscle expression [22]. The activating calcium for calcineurin may originate from numerous sources, but often, a transient receptor potential (TRP) family member, such as TRPC1, is involved in a variety of electrically excitable cell classes [23,24].

## 4. Mitochondrial Energy Status

The elevated respiratory activity associated with the undertaking of endurance exercise creates a deficit in mitochondrial respiratory co-factors that is manifested as increases in the ratios of AMP/ATP and NAD^+^/NADH. These alterations in energy status induce the expression of the transcriptional coactivator peroxisome-proliferator-activated receptor gamma coactivator-1 (PGC-1α), the master regulator of mitochondrial gene expression [28,29]. PGC-1α activity is post-translationally regulated by AMP-activated protein kinase (AMPK) phosphorylation (stimulated by a high AMP/ATP ratio) and NAD-dependent deacetylase Sirtuin–1 (Sirt1) deacetylation (stimulated by a high NAD^+^/NADH ratio) [28,30,31]. AMPK activation will also inhibit the mammalian target of rapamycin (mTOR) that will interfere with glycolytic-muscle determination [28,29]. Because of the elevated respiratory rates exhibited by oxidative muscles, muscle mitochondrial content and oxidative metabolism are enhanced, in conjunction with calcineurin signalling that consolidates the oxidative-muscle phenotype [32]. Therefore, it is the concomitant activation of these calcium and mitochondrial responses that establishes the genetic backdrop for the consolidation of the oxidative-muscle phenotype in response to exercise.

Intermediate subtypes within the glycolytic and oxidative phenotypes exist that are distinguished from each other based on protein isoforms comprising the contractile filaments. A detailed description of these contractile subtypes goes beyond the scope of this review and can be found elsewhere [22,33,34].

## 5. Secretome-Mediated Muscle–Fat Crosstalk

Due to their greater mitochondrial and myoglobin content and denser capillary beds, oxidative fibres are also sometimes described as “red”, whereas glycolytic fibres, for comparison, are often referred to as “white” [32]. Generally, exercise training produces a glycolytic to oxidative switch in muscle-fibre metabolism and contractile properties [33,34,35]. Analogously, adipose tissue can be broadly classified as either white or brown, where white adipose is specialised for energy storage and brown adipose is adapted for energy expenditure and thermogenesis. White adipose cells are larger, contain a single large unilocular lipid droplet and tend to be proinflammatory, especially in states of prolonged physical inactivity. Brown adipose cells, on the other hand, have multiple smaller lipid droplets and greater mitochondrial content. Brown adipose cells are, hence, smaller, darker and less inflammatory than white adipose cells [36,37]. The oxidative phenotypes of both oxidative muscle and brown adipose tissue are conferred by direct and indirect exercise-induced elevations in PGC-1α, respectively [32], and are mutually reinforced via an interplay between the muscle and adipose secretomes [38].

The muscle secretome is mobilised downstream of PGC-1α activation and hence, is enhanced by exercise [7,17]. Principal amongst the tissues targeted by the muscle secretome is adipose [38] and key amongst the adipose-regulating factors released by exercise is irisin [17,39,40] (Figure 2). Irisin is responsible for the induction of spontaneous energy expenditure by adipose tissue in the form of mitochondrial uncoupled respiration, otherwise known as adaptive thermogenesis [37,41], which attenuates systemic inflammation and promotes metabolic health. Irisin descriptively “beiges” white adipose to a browner phenotype [36,37]. Muscles, hence, get redder (more oxidative) in association with a “beiging” (more thermogenic) of collateral adipose depots. Although these changes are partially reversed following periods of sedentary behaviour, epigenetic responses to exercise governing systemic metabolism and longevity have been shown to persist long after training has stopped [42] and often implicate the PGC-1α-promoter region in humans [43] and mice [44]. The exercise response is most healthful when routinely reinforced.

PGC-1α is upregulated in adipose tissue [45,46] following its activation by irisin [17,39]. Irisin-activated adipose, in turn, reciprocally secretes adiponectin, which bolsters muscle and bone development and metabolism [47,48,49]. Brown adipose tissue (BAT) releases a distinct combination of adipokines from white adipose tissues, collectively known as batokines, that regulate the development and metabolism of muscle and collateral tissues [50]. Conversely, adiponectin is produced and secreted by skeletal muscle in response to exercise in association with oxidative-muscle expression [47]. Irisin, in turn, is produced and released by adipose in response to circulating irisin and exercise [39,51,52], acting to consolidate the exercise response via mutual secretome crosstalk. Irisin generally upregulates PGC-1α and the nuclear factor erythroid 2–related factor 2 (Nrf2) in recipient tissues [10]. In particular, the human visceral fat deposit has a preponderance to be highly inflamed and is a strong contributor to metabolic disruptions but is highly susceptible to exercise-induced browning [36]. Visceral fat, hence, represents a valid therapeutic target for the development of interventions to control the rising global incidence of metabolic dysfunction. Muscular respiratory activity, hence, elicits a systemwide cascade of PGC-1α-dependent secretome responses that underlie the metabolic benefits commonly attributed to exercise.

## 6. Fiber Type–Fiber Size Paradox

The signalling cascades elicited for either oxidative or glycolytic determination literally compete for biosynthetic priority. As a result, muscle size and oxidative capacity are inversely related. This occurs as a consequence of the balancing of myofibrillar and mitochondrial protein production, governing contractile and metabolic adaptations, respectively [53]. Specifically, heightened mitochondrial biogenesis downstream of AMPK and PGC-1α transcriptional cascades results in an overall attenuation of protein anabolism and smaller muscle fibre size. Indeed, muscle fibres with the highest oxidative capacity have been shown to have the smallest cross-sectional area [53]. A reduction in muscle fibre cross-sectional area is hypothesised to represent a metabolic adaptation with the objective of facilitating oxygen uptake into highly oxidative fibres by effectively increasing the fibre’s surface-to-volume ratio in association with heightened mitochondrial respiratory capacity and PGC-1α activity [53,54]. Cold-water immersion is a mode of water therapy that stimulates PGC-1α activity and it exhibits key aspects of this exercise–muscle size paradox [55]. Cold-water-immersion therapy post-exercise has been generally shown to diminish resistance-training adaptations, such as muscle hypertrophy, whilst aerobic exercise performance adaptations and secretome release have been shown to be enhanced [56]. Finally, calcineurin activation is not a prerequisite for the hypertrophic response of muscle [57]. Muscular calcineurin and mitochondrial activation of PGC-1α are, hence, associated with elevated muscle metabolic capacity and reduced muscle fibre size. Although exercise is the favoured approach to enhancing oxidative-muscle development, its adoption is difficult to implement in the old, frail and infirm (also see Section 12. *Is Magnetic Mitohormesis a Substitute for Exercise?*). Alternative modes of inducing the oxidative-muscle phenotype in these demographics are actively being sought to help curb the growing global prevalence of metabolic syndrome and to improve the quality of life in the swelling ranks of the advanced aged.

## 7. TRPC1 Promotes Oxidative-Muscle Development

The canonical transient receptor potential (TRPC) channel family exhibits a predilection for regulation by growth factors and is broadly involved in tissue development [58]. TRPC1 is the most ubiquitously expressed member of the family and is thought to act as a regulator of the other family members. Accordingly, TRPC1 has been implicated in diverse developmental programs, including that of oxidative muscle [19,59]. TRPC1 exhibits a capacity to heteromultimerise with the other TRPC family members, theoretically uniting their distinct activation modes into a single channel complex [60]. Mechanotransduction [60,61,62,63], magnetoreception [64,65,66,67,68], phototransduction [69,70,71], intracellular calcium sensing [72] and redox-sensing [73] are all parallel activation modes of TRPC1 that help subserve its role as an integrator of diverse forms of biophysical stimuli of consequence for development and disease [60,61,74], particularly with reference to skeletal muscle [75,76,77,78,79,80,81].

The weight of evidence indicates that TRPC1 is responsible for the elevated resting calcium levels that underlies oxidative-muscle determination. Exercise and mechanical loading promote oxidative-muscle expression, whereas physical inactivity or mechanical unloading result in oxidative-muscle loss. The elevated and sustained basal calcium levels that are required for oxidative-muscle development are diminished by physical inactivity and precede a reversal in oxidative character [21]. Paralleling the calcium signature of oxidative muscle, TRPC1 expression is highest in oxidative-muscle fibres [82] and wanes with disuse [19]. Accordingly, silencing TRPC1 expression results in oxidative-muscle loss [19]. TRPC1-mediated calcium entry activates the calcineurin/NFAT pathway [19,83] that, in turn, sustains TRPC1 transcription [19,84] as well as supporting PGC-1α function and oxidative-muscle maintenance [32]. Consequently, NFAT and TRPC1 levels decrease upon mechanical unloading and revert upon reloading, mirroring decreases and increases of oxidative-muscle expression, respectively [19]. Evidence thus supports cooperative roles for TRPC1, calcineurin/NFAT and PGC-1α in oxidative-muscle development, unified via mitochondrial respiration.

## 8. Magnetic Mitohormesis

Mitochondria can be considered the stress sensors of the cell and in this capacity serve to instil survival adaptations in response to oxidative stress produced during mitochondrial responses to environmental stimuli [85]. Mitohormesis refers to an adaptive process whereby low levels of reactive oxygen species (ROS) confer the installation of survival adaptations and promote regeneration, whereas greater levels of ROS can stymie cell growth and survival [86]. Mitohormesis is a manifestation of the ability of the mitochondria to either adapt to inherent oxidative stress by enhancing their anti-oxidant defences and improved respiratory efficiency or to succumb to oxidative damage before, or beyond, adaptations. Vital in the process of integrating mitohormetic adaptations are the PGC-1α (mitochondriogenesis) and Nrf2 (anti-oxidant defences) transcriptional pathways [28]. It has been noted that the concomitant increases in PGC-1α and Nrf2 expression and associated decrease in insulin/IGF signalling via mTOR signalling, resulting from endurance exercise, provide the appropriate oxidative milieu to establish mitohormetic survival adaptations [28,29]. As magnetic fields stimulate mitochondrial respiration, they can be exploited as a method with which to non-invasively produce mitohormetic responses, via a novel process of magnetic mitohormesis.

## 9. TRPC1 Confers Magnetic Mitohormesis

The enzymatic and genetic cascades activated by extremely low-frequency pulsed electromagnetic field (ELF-PEMF) exposure have recently come into clearer focus [87,88] and are commonly shown to invoke mitochondrial survival adaptations [89,90,91] and calcium signalling pathways [92,93,94,95,96,97] in a variety of cell classes. These two cellular response limbs need not be mutually exclusive from each other [65,90,91,98,99,100,101] but likely converge at the level of the mitochondria via the reciprocal and synergistic capacities of calcium to modulate mitochondrial respiration [102] and of mitochondrially-derived ROS to modulate the multimodal and integrative function of TRPC1 [73]. The manner in which the implicated calcium pathway interacts with the mitochondria most likely involves classical pathways [100], whereas the nature of magnetoreception in all likelihood entails magnetically-tuneable changes in the lifetime of a radical pair formed between a cryptochrome moiety and the mitochondrial cofactor, flavin adenine dinucleotide [87]; both mechanisms mutualistically reinforcing the other. Nonetheless, response to ELF-PEMFs has been recently shown to closely correlate with the developmental expression of TRPC1, whereas pharmacological inhibition or genetic silencing of TRPC1 precluded magnetoreception during in vitro myogenesis [65,68,102], chondrogenesis [64] and neurogenesis [67,103]. When specifically examined, these magnetically induced developmental responses invoke secretome activation [68,104]. Indeed, vesicular TRPC1 delivery was shown necessary and sufficient to reinstate magnetically-induced mitochondrial respiration and enhanced myogenesis in a CRISPR/Cas9 TRPC1-knockdown skeletal muscle cell line [66]. Finally, muscle cells that have been shielded from all ambient magnetic fields exhibited downregulated TRPC1 expression and reverted in vitro myogenesis [65]. Available evidence, hence, supports that TRPC1-mediated calcium entry is involved in transducing magnetic signals into diverse developmental responses by activating the calcineurin pathway (Figure 3).

## 10. Magnetic Mitohormesis Recapitulates Oxidative-Muscle Development in Mice

### 10.1. Magnetic Calcineurin/NFAT Induction

In agreement with previous findings [23,24], cyclosporin and TRPC1 channel antagonists were both capable of precluding ELF-PEMF-induced in vitro myogenesis, indicating that calcineurin was being activated by TRPC1-mediated calcium entry [65]. Brief ELF-PEMF exposure also preferentially enhanced the nuclear translocation of NFATC1, whereas NFATC3 nuclear localization was reduced [65], indicating a proclivity of myogenesis towards an oxidative phenotype [105]. Accordingly, oxidative-fibre expression was increased and was associated with a decrease in oxidative-fibre cross-sectional area in mice treated weekly with ELF-PEMFs and was accompanied by enhanced running performance [106]. A decrease in the cross-sectional area of oxidative-muscle fibres has previously been reported in response to myogenin overexpression [54] and has been shown to be inversely correlated with muscle-fibre oxygen consumption rate [53]. This morphological response is explained as a physical adaptation aimed at enhancing oxygen transport in highly oxidative fibres.

Provocatively, TRPC1 and mitochondria may reciprocally reinforce the response of the other via the capacity of calcium to modulate mitochondrial respiration [102] and of mitochondrially-derived ROS to modulate TRPC1 function [73]. It is intriguing to speculate that this TRPC1-mitochondrial nexus may represent a manner for the PGC-1α and calcineurin limbs of oxidative-muscle development to coordinate their developmental directives. In the exercise scenario, mechanical forces can activate TRPC1-mediated calcium entry (calcineurin induction) and central-nervous-system drive for movement will activate mitochondrial respiration (PGC-1α induction).

### 10.2. Magnetic Modulation of Mitochondrial Energy Status

Additionally, ELF-PEMF stimulation of muscle also was shown to activate mitochondrial respiration [65]. Accordingly, the developmental adaptations observed in response to ELF-PEMF stimulation closely mirrored those commonly attributed to endurance exercise downstream of PGC-1α activation. Brief (10 min) exposure of muscle cells to low energy ELF-PEMFs was demonstrated to stimulate in vitro myogenesis in association with increased mitochondrial number, heightened mitochondrial antioxidant defences and enhanced mitochondrial-based survival adaptations [65,68]. These effects were detected in isolated muscle cells [65] as well as intact muscle [106] and were associated with transcriptional activation of PGC-1α and Nrf2, previously shown to be involved in promoting mitochondriogenesis and resistance to oxidative stress [28,107], respectively. These same transcriptional pathways are also known to be activated by endurance exercise and moderate oxidative stress [10,108]. Consistent with an increased expression of PGC-1α, mice exposed to 1 mT PEMFs for 10 min per week for several weeks [106] exhibited metabolic and functional adaptations similar to those commonly attributed to oxidative-muscle development, including enhanced resting fatty-acid oxidation (reduced respiratory exchange ratio) [109], reduced resting insulin levels [110], enhanced running performance [109] and stimulated muscle mitochondrial fatty-acid transport [111,112].

## 11. Adipogenic Consequences of ELF-PEMF Stimulation

The adipogenic consequences of weekly ELF-PEMF treatment in mice were particularly robust and associated with elevated PGC-1α expression in both white and brown adipose samples [106]. In essence, ELF-PEMF treatment, in combination with treadmill running, increased PGC-1α most strongly in white adipose and was associated with an increase in uncoupled mitochondrial respiration (adaptive thermogenesis) and adipose mitochondrial content, as reflected by increases in the expressions of uncoupling protein 1 (*Ucp1*) and mitochondrial cytochrome c oxidase polypeptide 7A1 (*Cox7a1*), respectively [113]. These responses paralleled previously reported disparate thermogenic responses of the white versus brown adipose deposits to exercise [114]. The typically exercise-associated reddening of skeletal muscle and the browning of adipose tissue was therefore recreated in mice receiving brief weekly ELF-PEMF treatment (Figure 4).

The high caloric value of lipids makes them the best energy source to fuel the sustained physical activity commonly reserved for oxidative muscle. Therefore, stimulating muscular mitochondrial respiration and consequent PGC-1a expression, whether by exercise [115] or ELF-PEMF exposure [106], increases muscular mitochondrial fatty-acid oxidation and oxidative-muscle character. In an adaptive response aimed at making lipids more readily available as fuel to meet enhanced fatty-acid oxidation, intramyocellular lipid content increases in oxidative muscle in response to either training [116] or ELF-PEMF therapy [117] in humans. Oxidative muscle, hence, specifically sequesters lipids during periods of training to accommodate increased energy demand.

On the other hand, reduced physical activity results in the inappropriate ectopic accumulation of adipose tissue within muscle as well as extramuscular sites that is associated with increased serum levels of ceramides. Muscle ceramide levels are greater in sedentary individuals where they are strongly correlated with metabolic disturbances [118]. On the other hand, reducing skeletal muscle ceramide levels is associated with improvements in systemic insulin sensitivity [119], yet can occur independently of muscle growth [120]. Accordingly, elevated ceramide levels have been shown to undermine muscle maintenance and metabolism [121]. Importantly, certain long-chain ceramide species have been shown to be toxic to mitochondria [121,122]. Indeed, blood ceramide levels may be a more precise predictor of cardiovascular disease and diabetes than even cholesterol and may underscore an unmet need for ceramide-lowering therapeutics [122].

Oxidative fibres are enriched in lipids that are reduced during endurance exercise due to the enhanced oxidation of fatty acids by mitochondria [116]. Accordingly, ceramides accumulate within oxidative fibres during periods of physical inactivity [118] and are reduced by endurance exercise [123]. In alignment with the oxidative phenotype, muscular overexpression of PGC-1α in obese mice similarly reduced ceramide levels but increased mitochondria content and mitochondrial fatty-acid uptake [124]. Conversely, transgenic silencing of TRPC1 expression in mice was shown to elevate serum ceramide levels, corroborating a role for TRPC1 in cellular energy homeostasis [125]. These results clearly implicate oxidative muscle, and its propensity for fatty-acid oxidation, as a therapeutic target-tissue to control systemic ceramide levels. Notably, a recent human study has shown that brief (10 min) weekly ELF-PEMF exposure of the operated legs of patients (19–42 years of age) after having undergone anterior cruciate ligament reconstructive surgery produced significant reductions in serum ceramide levels compared to a sham control cohort [117]. The intervention period was 16 weeks and additionally produced indications of improved muscle regeneration and improved systemic metabolic status.

Age-related reductions in physical activity aggravate white adipose inflammation and promote its redistribution to visceral and ectopic intramuscular sites. Intramuscular atherogenic adipose accumulation augments ceramide production, resulting in mitochondrial dysfunction and enhanced oxidative stress. The reduction in mitochondrial efficiency depresses fatty-acid β-oxidation, which further exacerbates intramuscular lipid accumulation and ultimately leads to insulin resistance as well as accelerating muscle weakness and atrophy. Under these debilitating conditions the muscle secretome switches to a more inflammatory status, upregulating the secretion of pro-inflammatory myokines (e.g., TNF-α and IL-6), while reducing the secretion of anti-inflammatory myokines (e.g., irisin and adiponectin). This shift in the muscle secretome towards one that is more pro-inflammatory, in turn, stimulates the release of pro-inflammatory adipokines and immune cell cytokines, which further aggravates adipose and system-wide inflammation, setting into motion a vicious cycle of metabolic and functional decline that characterise the pathogenesis of sarcopenia [126,127,128,129] (also see Section 14, *Magnetic Mitohormetic Implications for Lifespan*). The systemic inflammatory milieu that results from the sarcopenic condition is hypersensitive to even minor environmental stressors and plays a major role in the pathophysiology of age-related frailty in humans [130]. Decisive in the progression of this deteriorating metabolic scenario is a pro-inflammatory shift in the myokine–adipokine interactions due to mitochondrial dysfunction and largely downstream of PGC-1α transcriptional deficiency.

Notably, asprosin is a pro-inflammatory adipokine secreted by white adipose tissue. Adenovirus-induced overexpression of asprosin in subcutaneous white adipose was shown to reduce adaptive thermogenesis as well as decrease the expressions of UCP1 and PGC-1α as well as other browning-related genes, while upregulating the expression of genes associated with general adipogenesis. Asprosin overexpression in mice also suppressed cold-induction of Nrf2. In vitro, adenovirus-mediated overexpression of asprosin in primary adipocytes inhibited adipose browning and aggravated lipid deposition and could be reversed with the Nrf2 agonist, oltipraz. Asprosin, hence, obstructs browning and promotes lipid deposition in adipose tissue via a Nrf2-mediated mechanism [131].

Visceral adiposity is particularly inflammatory in nature [36]. Visceral fat is correlated with elevated serum ceramide levels and insulin-resistance [132]. Another recent study showed significant reductions in total and visceral fat deposits in a community-based study examining an older (38–91 years of age) cohort of volunteers. Strikingly, reductions in intra-abdominal and total fat were observed after eight weeks of ELF-PEMF exposure, representing only 80 min of total exposure (10 min/week) [133]. This finding aligns with previous data showing that visceral fat is particularly prone to exercise-induced adaptive thermogenesis [36,134]. These adipose changes were accompanied by improved functional mobility as indicated by the Timed Up and Go, Five Times Sit-to-Stand, and 4 m Normal Gait Speed tests as well as significant increases in lean muscle mass as determined by bioelectrical impedance analysis, particularly in the more elderly of the cohort. These data support the notion that human ELF-PEMF therapy may represent a method to recapitulate a subset of the metabolic benefits commonly associated with endurance exercise in a simple and non-invasive therapeutic platform and is supported by a growing body of preclinical studies. These benefits would hold particular importance in the frail and elderly that would otherwise be incapable of undertaking exercise due to weakness.

At the systemic level, the adipogenic consequences of ELF-PEMF therapy were more pronounced than the muscular effects [65,106,133]. These adipogenic effects were clearly apparent in mice despite both white and brown adipose cells in vitro not being responsive to the same ELF-PEMF signature applied to the animal, and shown to be effective on muscle both in vitro and in vivo [65,106]. Studies focussed on endurance exercise agree with the results generated in response to magnetic mitohormetic strategies, demonstrating clear indices of metabolic improvement, whereas muscle mass was modestly improved [117,133]. Nonetheless, improvements in functional mobility were observed in a frail cohort receiving ELF-PEMF therapy, likely reflecting increases in oxidative muscular capacity, enhanced resistance to fatigue and improved metabolism [133].

## 12. Is Magnetic Mitohormesis a Substitute for Exercise?

Resistance exercise, and endurance exercise to a lesser degree, produce mechanical stress and microdamage, which may serve as a stimulus to spur muscle regrowth [135,136]. In this respect, physical exercise also creates a biosynthetic regenerative energy debt that will need to be allocated towards the repair and rebuilding of damaged muscle tissue (Figure 5). Inadvertently, this will limit biosynthetic energy availability to implement metabolic adaptations. ELF-PEMF exposure, as it is low energy and non-mechanical, represents a more focussed form of mitochondrial stimulation. ELF-PEMF exposure may thus render stronger mitochondrial responses via the PCG-1α and Nrf2 transcriptional pathways, while producing relatively less impetus for muscle hypertrophic remodelling. It would thus be better to combine PEMF therapy with exercise, if and when possible, for greatest physiological synergism. On the other hand, mechanical stress may need to be avoided by certain frail patient demographics that, by necessity, refrain from physical exercise and consequently suffer metabolic disruption. Magnetic therapy may serve as an option for such frail demographics to maintain metabolic balance and to exploit the systemic regenerative benefits conferred by the muscle secretome. Pain and weakness are strong deterrents to exercise in the elderly [16]. Because of physical inactivity arising from such inflammatory circumstances the elder community commonly suffers from metabolic dysfunction and physical frailty. It was shown that functional mobility improved in conjunction with reduced pain in a group of elderly subjects receiving weekly ELF-PEMF treatment [133]. These results suggest that magnetic therapy may represent one manner to capacitate the elderly to undertake exercise more readily. In support of potential magnetic-therapy–exercise synergism, it was previously shown that mice, having received ELF-PEMF treatment, exhibited improved running performance after five weeks (10 min of exposure per week) [106].

## 13. Health Implications of Muscle-Targeted Magnetic Mitohormesis

Human health is intimately linked to muscle health, which improves with exercise, as do healthspan and longevity. The ability of muscle to adapt to exercise is mitochondria-dependent. Disruptions in both calcineurin [22,137] and PGC-1α [138] signalling negatively impact oxidative-muscle development with understandable negative ramifications over metabolism and disease. Evidence for mechanistic synergism between the calcineurin and PGC-1α pathways arises from the finding that calcineurin knock-down alters mitochondrial turnover and respiration, attenuates exercise capacity, and disrupts adipose energy storage [139]. Accordingly, the calcineurin and PGC-1α pathways are jointly responsible for oxidative-muscle determination [32]. Oxidative muscle is characterised by elevated mitochondrial content and high reliance on fatty-acid oxidative metabolism [32]. Oxidative muscle, hence, exhibits a predilection for fatty-acid oxidation that supports insulin-sensitivity [20]. The adipogenic consequences of muscle magnetic therapy recapitulates several metabolic features typically attributed to endurance exercise and thus has far-reaching health implications arising from its proclivity to efficiently promote adipose browning [106], reduce serum ceramides [117] and reduce total and visceral fat [36,133], which should serve to ameliorate human metabolic and inflammatory disorders, such as COVID-19 [85].

### Duchenne and Becker Muscular Dystrophy

Duchenne muscular dystrophy (DMD) is an X-linked, muscle-wasting disease, affecting approximately one in every 3500–4000 newborn males worldwide, making it the most common form of muscular dystrophy [140]. DMD results from loss-of-function mutations in the dystrophin gene, encoding for the dystrophin protein that is essential for maintaining the structural integrity of the muscle surface membrane [141]. The muscles of individuals inflicted with DMD are, hence, highly susceptible to mechanical damage, leading to premature death [142]. Becker muscular dystrophy (BMD) is a milder version of the disease and arises from dystrophin mutations that lower dystrophin expression or lead to the accumulation of an internally truncated dystrophin protein that reduce functional capacity. Dystrophin deficiency leads to progressive skeletal muscle-wasting accompanied by increased inflammatory adipose infiltration and fibrosis. Clinical strategies proposed to restore dystrophin in muscle include various forms of gene and stem-cell therapies, which would switch DMD patients to a more BMD-like condition, thus lowering the risk of muscle-wasting and achieving a longer lifespan. Gene therapies aimed at reinstating dystrophin face major challenges due to its large size (427 kDa) and the requirement for the effective delivery of the replacement gene to the body’s largest tissue mass, skeletal muscle [143]. 

A more practical approach would be to catalytically upregulate the expression of existing dystrophin homologues. Employing the *mdx* mouse model of human BMD, it was shown that stimulating calcineurin signalling upregulated the expression of utrophin, the autosomal homologue of dystrophin, and was associated with ameliorated muscle damage [144]. Moreover, overexpressing calcineurin in *mdx* skeletal muscle decreased muscle pathology as well as increased the expressions of both utrophin and oxidative fibres [145], aligning with previous evidence indicating that glycolytic fibres are more vulnerable to the condition [22]. The preferential calcineurin activation elicited by targeted ELF-PEMF therapy may, hence, be of clinical consequence in the realm of DMD and BMD. Furthermore, calcineurin activity in DMD is also compromised by the prevalent oxidative stress characteristic of the condition [146]. In this regard, magnetic mitohormesis may also be beneficial in reestablishing oxidative balance and in reinstating calcineurin function.

## 14. Magnetic Mitohormetic Implications for Lifespan

A biosynthetic penalty seems to exist with reference to lifespan. Its roots extend from the finding that inhibiting the mTOR pathway, which governs protein accrual and is, hence, responsible for muscle hypertrophy, with rapamycin ameliorates immunosenescence [147] and extends lifespan [148]. On the other hand, insulin and the insulin-like growth factor 1 (IGF-1) are potent activators of mTOR and muscle growth [149] and are negatively correlated with longevity [150]. It is broadly recognised that mitochondrial health and human health are strongly intertwined and, conversely, that ageing is associated with impaired mitochondrial maintenance [151]. Notably, the promotion of oxidative-muscle development by PGC-1α comes at the expense of mTOR activation and muscle hypertrophy in favour of mitochondrial pathways [28,35,53]. Activation of the AMPK/PGC-1α pathway extends lifespan while inhibiting the insulin/Akt/mTOR pathway [28,152]. Evidence also exists that caloric restriction extends lifespan [153,154,155]. As caloric restriction shares many of the hallmark features of our magnetic-stimulation paradigm such as Sirt1 and PGC-1α activation, upstream of mitochondrial biogenesis [65,106], muscle magnetic-field therapy may offer a non-invasive method to promote longevity. Indeed, Sirt1 and PGC-1α activities converge at the level of mitochondrial respiration. In response to mitochondrial low-energy status, manifested by an elevated NAD^+^/NADH ratio, Sirt1 deacetylates PGC-1α, rendering it capable of promoting mitochondriogenesis [31]. It is thus intriguing to speculate that attenuating mTOR-mediated biosynthesis via the magnetic induction of the calcineurin, PGC-1α and Sirt1 pathways, may serve to help improve the quality of life of the elderly. Finally, the muscle secretome has recently been shown to contain factors, such as the DEP-domain-containing mTOR-interacting protein (DEPTOR), that specifically inhibits mTOR [156]. Magnetic therapeutic interventions may potentially provide a manner to shift the relative emphasis of the disparate glycolytic and oxidative-muscle secretomes [157] for the modulation of human healthspan and lifespan.

### Sarcopenia

Sarcopenia is the age-related loss of muscle and physical decline that plague the elderly [158]. A selective loss of glycolytic-muscle fibres and their skeletal muscle satellite cell pool characterise sarcopenia, whereas the number of oxidative-fibre-associated satellite cells was similar with age [159]. Specifically, sarcopenic glycolytic fibres exhibit a pronounced loss in cross-sectional area (~26%), whereas the cross-sectional area of oxidative fibres was unchanged relative to the young [160]. Mitochondrial dysfunction was also more prevalent in glycolytic muscle in ageing mice but was maintained in oxidative muscle [161]. In an intriguing parallel to DMD, oxidative fibres exhibit superior survival potential. One interpretation of the available evidence is that oxidative muscle confers survival to the individual during age-related muscle loss, reflecting frailty merely by association, and may, hence, represent a viable target for therapeutic intervention. Finally, a loss in muscle mechanosensitivity contributes to the aetiology of sarcopenia [162]. The pertinent question, hence, becomes whether sarcopenic muscle maintains developmental sensitivity to magnetic-field therapy—which could be exploited as a clinical intervention.

For the sake of brevity, detailed descriptions of the plethora of components that comprise the muscle and adipose secretomes, as well as the diverse subtypes of muscle and adipose, were excluded from the present review. 

## 15. Conclusions

Magnetic mitohormesis offers a novel way to recapitulate some of the metabolic responses commonly associated with the undertaking of endurance exercise, yet with a minimum of mechanical stress. Common to both muscle-targeted magnetic mitohormesis and endurance exercise is the activation of the PGC-1α transcriptional pathway. It is widely agreed that aerobic-exercise-induced enhancements in muscle PGC-1α expression sets off a chain of events that that improve systemic metabolic balance, combat disease and infection, reduce frailty, improve cognitive function and extend lifespan. Nonetheless, exercise is more encompassing, albeit more difficult to undertake, than low-energy magnetic-field therapy. For instance, physical exercise improves central-nervous-system communication with muscles, a process called motor learning, and increases blood circulation and heart rate as well as sympathetic drive and physiological responses that will not be immediately altered by magnetic-field therapy. On the other hand, magnetic-field therapy may provide a viable option for muscle and metabolic maintenance, or improvement, in the elderly and frail with limited capacity for exercise, which may ultimately lead to increased independence and physical capacity in these individuals, as supported by existing studies. Further work is required to elucidate the unique attributes as well as trade-offs of magnetic-field therapy relative to endurance exercise. To address these unknowns, additional studies examining the effects of brief and non-invasive low energy magnetic-field therapy in the areas of metabolic stabilisation and extended lifespan in humans are warranted.

## Figures and Tables

**Figure 1 bioengineering-10-00956-f001:**
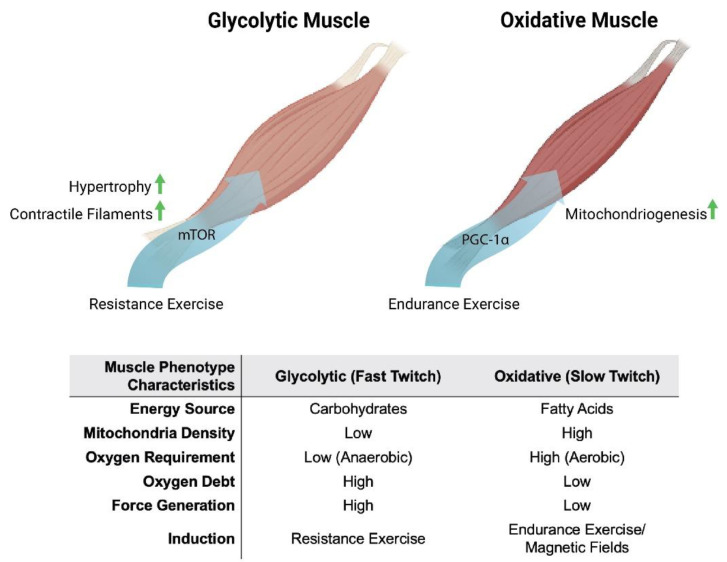
Metabolic and phenotypic characteristics of glycolytic and oxidative muscles in response to exercise and magnetic fields. Original figure created with BioRender.com.

**Figure 2 bioengineering-10-00956-f002:**
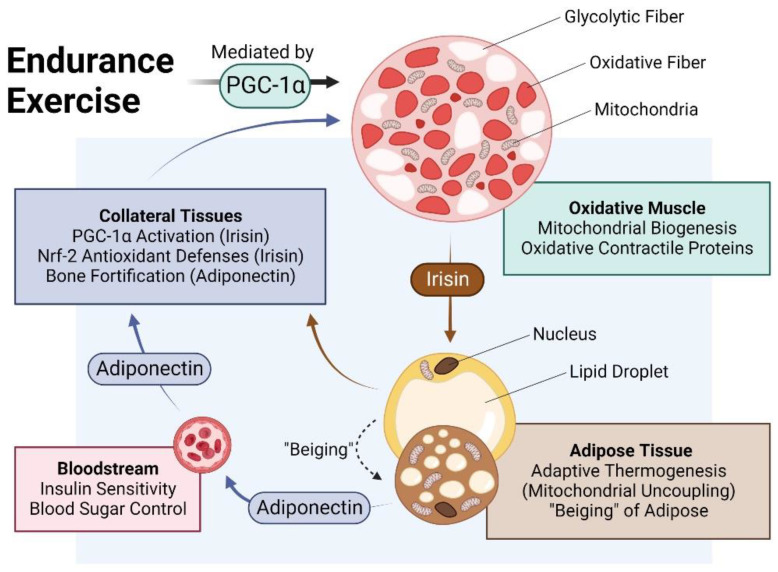
Impact of endurance exercise on the crosstalk between oxidative muscle and adipose tissue. Original figure created with BioRender.com.

**Figure 3 bioengineering-10-00956-f003:**
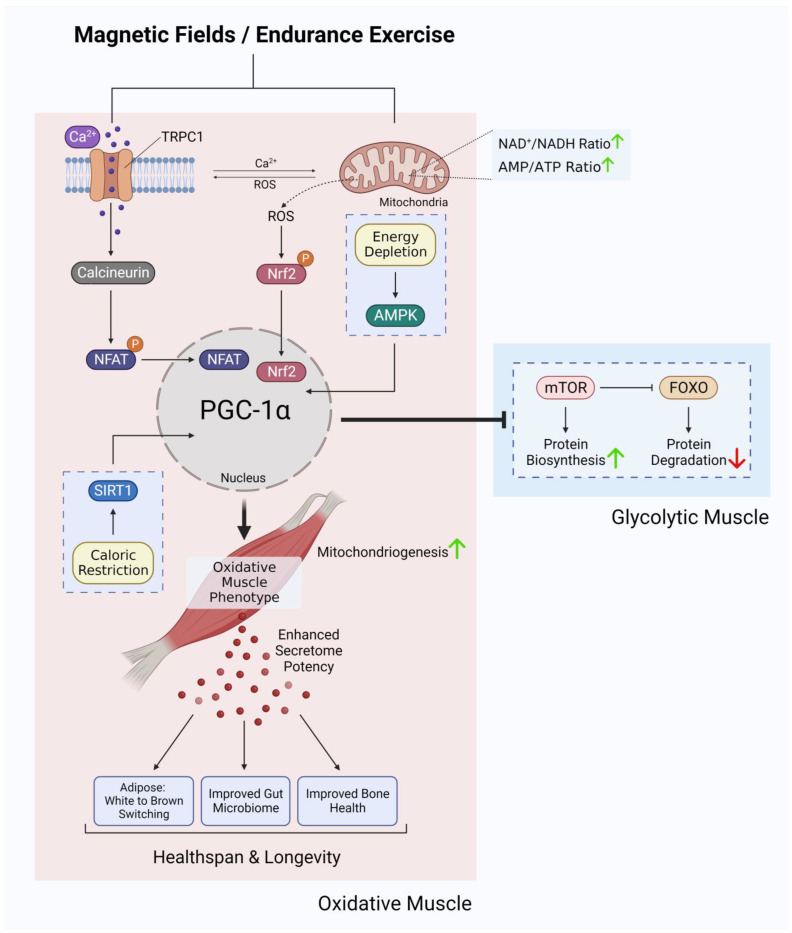
Ca^2+^ signalling and mitochondrial adaptations shared between magnetic fields and endurance training in the development of oxidative muscles. Original figure created with BioRender.com.

**Figure 4 bioengineering-10-00956-f004:**
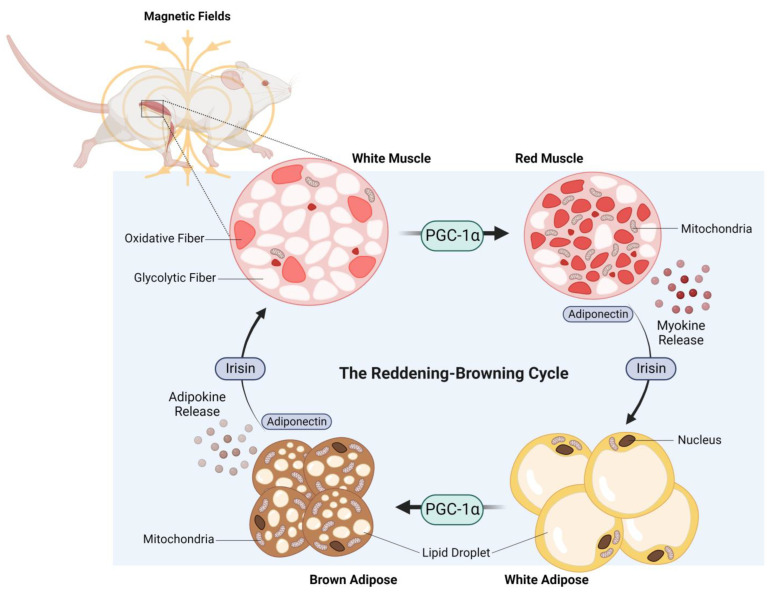
The reddening–browning cycle of magnetically-induced oxidative-muscle development closely mirrors the endurance-exercise-based metabolic adaptations. Original figure created with BioRender.com.

**Figure 5 bioengineering-10-00956-f005:**
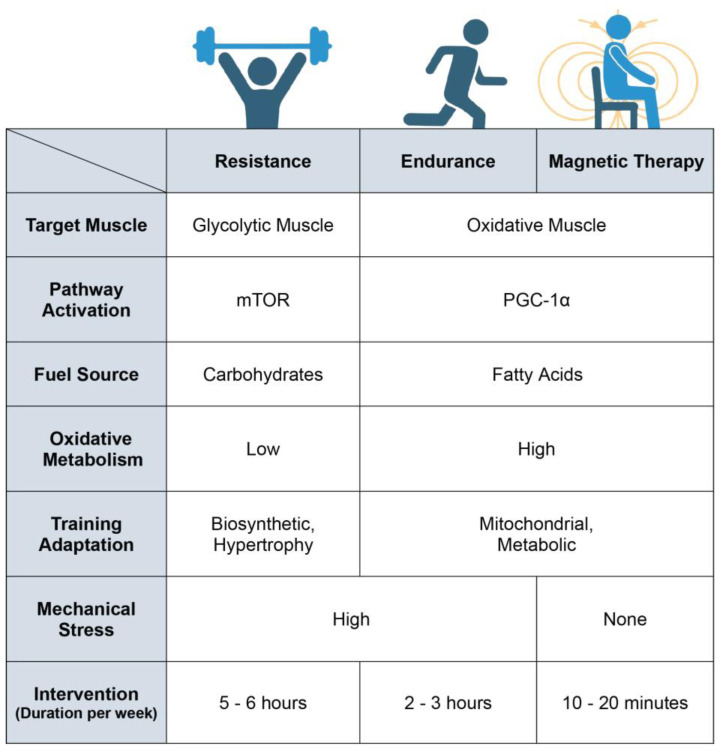
Breakdown of the physiological differences between resistance training, endurance training and magnetic-field intervention. Magnetic therapy refers to the use of extremely low frequency pulsed electromagnetic fields (ELF-PEMFs) such as those previously described [64,65,66,67,68,104,106,117,133]. Original figure created with BioRender.com.

## Data Availability

Not applicable.

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
