# Peer review of "The Developmental Implications of Muscle-Targeted Magnetic Mitohormesis: A Human Health and Longevity Perspective"

_bioengineering, 2023, doi:10.3390/bioengineering10080956_

Round 1

Reviewer 1 Report

Please discuss more the primary target or interface of the magnetic pulses and the membrane receptors on the mitochondrial membrane. In principle the TRPC1 receptor is electro.insensitive. However it cooperates with Ca channels or at least via Ca++. Could it be that voltage dependent Ca-channels are involved?

Only minor flaws in English language - final editing by native speaker suggested.

Reviewer 2 Report

In this paper, Franco-Obregon and colleagues advocate the use of electromagnetic fields to stimulate healthy muscle function in the context of specific pathological states, as well as in the face of normal human aging.  Overall, they provide an excellent overview of the basic muscle physiology with an emphasis on mitochondrial function, the modifying effect of exercise on these pathways and the parallels between exercise and electromagnetic therapy at the cellular and biochemical level.

In the section on the third full page of the body of the text, there is a thorough description of the adipose/muscle secretomes and how they influence one another, but I think including a figure here could be helpful.

On the following page there is reference to exercise being difficult in the elderly. And although it is probably self-evident for most, I would recommend expanding on this just a little with specific examples.

Finally, a summary table that compared the results of various “classic” interventions with electromagnetic stimulation side-by-side would be useful.  Although everything is described adequately in the text, there are a number of parts to keep track of.
